# pH-Responsive Lipid Nanocapsules: A Promising Strategy for Improved Resistant Melanoma Cell Internalization

**DOI:** 10.3390/cancers13092028

**Published:** 2021-04-22

**Authors:** Vincent Pautu, Elise Lepeltier, Adélie Mellinger, Jérémie Riou, Antoine Debuigne, Christine Jérôme, Nicolas Clere, Catherine Passirani

**Affiliations:** 1Micro & Nanomedecines Translationnelles (MINT), University of Angers, Inserm, The National Center for Scientific Research (CNRS), SFR ICAT, F-49000 Angers, France; v.pautu@gmail.com (V.P.); elise.lepeltier@univ-angers.fr (E.L.); adelie.mellinger@univ-angers.fr (A.M.); jeremie.riou@univ-angers.fr (J.R.); nicolas.clere@univ-angers.fr (N.C.); 2Center for Education and Research on Macromolecules (CERM), Complex and Entangled Systems from Atoms to Materials Research Unit (CESAM-RU), University of Liège, 4000 Liège, Belgium; adebuigne@uliege.be (A.D.); c.jerome@uliege.be (C.J.)

**Keywords:** nanomedicine, pH-sensitive, MDR cancer, tumor cell internalization, N-vinylpyrrolidone, vinylimidazole

## Abstract

**Simple Summary:**

Only 13 to 50% of patients with metastatic melanoma respond to new commercialized therapies. The reason why the same chemotherapeutic treatments yield different responses in patients can be attributed to the degree of multidrug resistance (MDR) developed by the host tumor cells. For instance, the glycolytic metabolism of cancer cells enhances the intratumoral accumulation of lactic acid, decreases intratumoral pH and potentiates MDR. Lipid nanoparticles (LNC) have been widely exploited as carriers of MDR reversing molecules. In this study, we proposed to modify LNC with novel copolymers to impart stealth properties and to improve tumor cell entry. Modified-LNC showed in vitro pH-responsive properties characterized by an enhanced cellular uptake under acidic conditions. Moreover, surface modification led to an increased biological effect by protecting the nanocarrier from opsonization by complement activation. These data suggest that pH-sensitive LNC are promising nanocarriers to target metastatic melanoma.

**Abstract:**

Despite significant advances in melanoma therapy, low response rates and multidrug resistance (MDR) have been described, reducing the anticancer efficacy of the administered molecules. Among the causes to explain these resistances, the decreased intratumoral pH is known to potentiate MDR and to reduce the sensitivity to anticancer molecules. Nanomedicines have been widely exploited as the carriers of MDR reversing molecules. Lipid nanocapsules (LNC) are nanoparticles that have already demonstrated their ability to improve cancer treatment. Here, LNC were modified with novel copolymers that combine N-vinylpyrrolidone (NVP) to impart stealth properties and vinyl imidazole (Vim), providing pH-responsive ability to address classical chemoresistance by improving tumor cell entry. These copolymers could be post-inserted at the LNC surface, leading to the property of going from neutral charge under physiological pH to positive charge under acidic conditions. LNC modified with polymer P5 (C18H37-P(NVP21-co-Vim15)) showed in vitro pH-responsive properties characterized by an enhanced cellular uptake under acidic conditions. Moreover, P5 surface modification led to an increased biological effect by protecting the nanocarrier from opsonization by complement activation. These data suggest that pH-sensitive LNC responds to what is expected from a promising nanocarrier to target metastatic melanoma.

## 1. Introduction

In recent years, cancer treatments have evolved considerably with the rise of new strategies for targeting the tumor microenvironment. Thus, targeted therapies and monoclonal antibodies have improved the prognosis and survival of many cancer patients. Despite these significant advances, low response rates and multidrug resistance (MDR) have been described for certain molecules, reducing their efficacy. For instance, in metastatic melanoma, it has been reported that only 13 to 50% of patients respond to these new therapies with, sometimes, a delay of more than three months [1,2]. The reason why the same chemotherapeutic treatment yields different responses in patients can be attributed to the degree of resistance developed by the host tumor cells [3].

MDR is associated with a wide range of pathological changes at different cellular and tissular levels. In a previous study, we found some differences in B16F10 and SKMel28 melanoma vasculature that could potentiate therapy resistance [4]. Moreover, MDR can be associated with somatic mutations. For instance, the most recurrent ones in chronic sun-induced damage (CSD) and non-CSD melanoma affect genes in key signaling pathways involved in proliferation (BRAF, NRAS and NF1), growth and metabolism (PTEN and KIT), cell identity (AT-rich interaction domain 2 (ARID2)), resistance to apoptosis (TP53), cell cycle control (cyclin-dependent kinase inhibitor 2A (CDKN2A), and replicative lifespan (telomerase reverse transcriptase (TERT)) [5].

Among the various hypotheses explaining MDR, recent data have suggested that the tumor microenvironment could play a key role in this therapeutic resistance [6]. The originality of the tumor microenvironment is based on its composition with various cells, such as endothelial or immune cells and cancer-associated fibroblasts, which may be involved in MDR [4,7]. However, many other physiological characteristics (hypoxia, acidity,…) can modify both tumor therapy and tumor progression [8,9]. Thus, changes in the pH of the tumor microenvironment can modify the efficacy of several anticancer chemotherapies. In fact, the glycolytic metabolism of cancer cells potentiates the intratumoral accumulation of lactic acid, decreases intratumoral pH [10] and potentiates MDR by reducing the sensitivity of tumor cells to anticancer molecules [11].

Nanomedicines have been widely exploited as the carriers of MDR reversing molecules [12]. Moreover, when properly formulated in terms of chemical composition and physicochemical properties, nanomedicines in themselves can help overcome MDR even without carrying a load of chemosensitizers [3]. Among them, lipid nanocapsules (LNC) can offer interesting perspectives for MDR cancer treatment because they have intrinsic characteristics, which are perfectly poised to address classical chemoresistance [13]. Based on the phase inversion process [14], LNC formulation is solvent-free and produces nanoobjects with a monodisperse and monitor size. Their structure is intermediate between that of polymer nanocapsules and liposomes and similar to that of lipoproteins because of their oily core, which is surrounded by a surface-active membrane of polyethylene glycol (PEG, 660 Da). Interestingly, this surfactant can inhibit drug efflux through inhibition of *P*-gp as it has been reported with etoposide- or paclitaxel-loaded lipid nanoparticles in in vitro and in vivo studies [15,16,17,18]. This original and groundbreaking work on LNC [15,16] has paved the way for other works showing the efficacy of ferrocifen-loaded LNC against MDR tumors [19,20,21,22,23,24,25,26], especially in malignant glioma models.

These last years, much progress has been motivated in *stimuli*-responsive nanocarriers, which could adapt to the intrinsic physicochemical and pathological factors to increase the specificity of drug delivery. Currently, numerous nanocarriers have been engineered with physicochemical changes in responding to external *stimuli*, such as pH, due to the nature of low pH inside the organelles (e.g., lysosomes and endosomes) of cancer cells and in the tumor microenvironment, a way to overcome multidrug resistance. Indeed, positively charged nanocarriers generally exhibit a higher cell uptake due to increased electrostatic interactions between engineered nanomaterials and the negatively charged cell membrane [27,28,29]. Therefore, considering this beneficial surface charge effect and the acidification of the tumor microenvironment caused by glycolysis metabolism of tumor cells [30], pH-responsive nanocarriers have been developed, allowing to increase cellular uptake and/or trigger the drug release into the tumor environment. Several polymers have been proposed as protonation agents to create these pH-responsive carriers [31,32] as vinyl imidazole (Vim) used for biomedical application purposes [33,34].

In this study, poly(N-vinylpyrrolidone) (PNVP) [35], which has been reported to prolong the circulation time of nanocarriers, was randomly copolymerized with Vim (data not shown). Thus, a new class of copolymers was developed by specific RAFT polymerization [36], combining the promising stealth properties of PNVP with a pH-responsive ability due to Vim and end-capped by a lipophilic C_18_H_37_- alkyl chain able to be post-inserted at the surface of LNC. A library of copolymers of various compositions and molar masses was then used to decorate LNC (Figure 1), and the physicochemical characteristics of the modified LNC were studied. Afterward, the impact of such decoration on in vitro cell toxicity was evaluated. Furthermore, the cell uptake of modified LNC was monitored on two melanoma cell lines (B16F10 and SKMel28) in a range of pH values. Finally, in acidic conditions, the internalization pathways of these nanovectors were assessed in these two cellular models.

## 2. Material and Methods

### 2.1. Formulation and Characterization of Nanoparticles

#### 2.1.1. Blank Formulation of LNC (BLK)

Lipid nanocapsules were formulated as described [14] by mixing under magnetic stirring 20.2% *w/w* caprylic-capric acid triglycerides (Labrafac WL 1349, Gatefossé S.A., France), 17.2% *w/w* hydroxystearate-PEG (Kolliphor^®^ HS 15, BASF, Ludwigshafen, Germany), 8% *w/w* NaCl (Prolabo, France), 1.5% *w/w* lecithin (Lipoid S75-3, Lipoid GmbH, Ludwigshafen, Germany) and 35.58% of Milli-Q water. Three cycles of heating and cooling between 60 and 90 °C were carried out to obtain the phase inversion. Ice-cold Milli-Q water 23.72% *w/w* was added at the last inversion temperature leading to the formation of lipid nanocapsules.

#### 2.1.2. Fluorescent LNC Formulation

To formulate fluorescent LNC, a fluorescent probe, DIA (4-(4-(dihexadecylamino)styryl)-N-methylpyridinium iodide; λ_excitation_ = 456 nm; λ_emission_ = 590 nm, Fisher Scientific, Illkirch-Graffenstaden, France), was used. Prior to formulation, DIA was incorporated in Labrafac at a concentration of 7.92 mg/g, then the same formulation process as above was used to formulate LNC.

#### 2.1.3. Polymer Post-Insertion

LNCs and polymers from Table 1 were coincubated for 2 h at 60 °C under magnetic stirring with a final polymer concentration of 1 mM. The mixture was then cooled down for 20 min in an ice bath under magnetic stirring. As a control, to obtain unmodified LNC (BLK), the same protocol was applied using the same dilution factor without polymer solution.

#### 2.1.4. Size and Zeta Potential Measurements

The mean diameter (Z-average) and polydispersity index (PDI) of LNC were measured using the dynamic light scattering (DLS) method at a backscatter angle of 173°. Zeta potential was measured using the laser Doppler microelectrophoresis technique with a Malvern Zetasizer^®^ apparatus (Nano Series ZS, Malvern Instruments S.A., Malvern, UK) at 25 °C. Measurements were performed in 1 mM phosphate buffer Na_2_HPO_4_ (Merk, Darmstadt, Germany) + NaH_2_PO_4_ (Sigma-Aldrich, Schnelldorf, Germany) at different pH: 7.4, 6.8, 6.5, 6. LNC was 10 times diluted. Experiments were conducted 4 times with a measured average value calculated from 3 runs, with 12 measurements by a run for size and 20 measurements by run for zeta potential.

#### 2.1.5. Stability

LNC was kept at 4 °C, stability in size, PDI, and zeta potential evaluated at days 0, 7, 14, 21. Experiments were repeated 3 times using the method described above.

#### 2.1.6. Endosome Buffering Effect

The buffering capacities of the different polymers alone and post-inserted into LNC were measured by acid–base titration according to the method described by Zhong et al. [37]. 1 mL of 0.1 mM polymer solution and 1 mL of post-inserted LNC was adjusted initially to pH 11 with 0.1 M NaOH. Then, the solutions were titrated to reach pH 3 by adding 0.1 M HCl. After each addition of 25 µL of HCl, pH was measured with a pH meter. Endosome buffering capacity was evaluated as the HCl amount to modify the solution pH from 7.4 to 5.1.

### 2.2. Cell Culture

The SK-Mel 28 human melanoma cell line, obtained from ATCC (LGC Promochem, France), and B16F10 mice melanoma cell line (gift from University of Brussels) were grown in Roswell Park Memorial Institute (RPMI) 1640 medium (Lonza, Verviers, Belgium) supplemented with 10% heat-inactivated FBS (Lonza), 1% antibiotic and antimycotic solution (Sigma-Aldrich) and 1% non-essential amino acids (Lonza). Cell lines were cultured and maintained at 37 °C in a humidified atmosphere with 5% CO_2_. To obtain pH-modified media, a complete culture medium was maintained for 24 h at 37 °C in a humidified atmosphere with 5% CO_2_. Media was then buffered to pH 7.4; 6.8; 6.5, and 6 using a NaH_2_PO_4_ 2 M solution.

### 2.3. Resazurin Cell Viability Assay

B16F10 and SK-Mel 28 cells were cultured in a 96-well plate at the density of 5 × 10^4^ cells/well and incubated for 24 h. Cells were then treated with 100 µL/well of media containing either post-inserted LNCs at a concentration range of 10 µg/mL to 1000 µg/mL or polymers at the corresponding concentration contained in post-inserted LNCs (0.025 µM to 2.5 µM). After treatment, cells were washed with PBS. Cytotoxicity was determined by evaluating cell viability via indirect quantification of living cells using resazurin assay (Sigma-Aldrich). Briefly, 100 µL of resazurin (22.5 µg/mL) was added to each well, and the plates were incubated for 3 h under a humidified atmosphere with 5% CO_2_. The fluorescence intensity was then measured at 544 nm excitation/599 nm emission wavelengths using a Fluoroskan Ascent™ microplate fluorometer (Fisher Scientific). Experiments were repeated six times with three replicates per experiment.

### 2.4. Fluorescence-Activated Cell Sorting (FACS): Internalization

The uptake of unmodified LNC (BLK LNC) and modified LNC at different pH was measured by flow cytometry analysis. Cells were seeded onto 6-well plates at the density of 5 × 10^6^ and 2.50 × 10^6^ cells/well for B16F10 and SK-Mel 28, respectively. Then, cells were incubated for 24 h and were then treated for 2 h with 2 mL medium at different pH (7.4, 6.8, 6.5, 6) with fluorescent LNCs at a concentration of 250 µg/mL. Cells were washed with PBS, collected after trypsinization and washed twice with PBS. To discriminate cell membrane adsorbed or internalized LNCs, cells were resuspended in 2.5 mg/mL trypan blue to quench extracellular fluorescence. Analyses were performed with a BD FACSCanto™ II flow cytometer (BD Bioscience, San Jose, CA, USA). Experiments were repeated four times with three replicates per experiment.

### 2.5. Fluorescence-Activated Cell Sorting (FACS): Internalization Pathway

To study the internalization pathways of pH-responsive LNC, different inhibitors were used. Cells were seeded onto 6-well plates at the density of 5 × 10^6^ cells/well for B16F10 and 2.5 × 10^6^ for SK-Mel 28 and incubated for 24 h. The cells were then treated with inhibitors for 30 min at 37 °C. Then, 1 mmol/L 5-(*N,N*-dimethyl)amiloride hydrochloride (DAM), 10 mg/mL methyl-β-cyclodextrin (MβC) and 50 µmol/L chlorpromazine (chlorpr.) were used to inhibit macropinocytosis, lipid RAFT and mediated endocytosis pathways, respectively [38]. 10 µg/mL phorbol 12-myristate (PMA) (Sigma-Aldrich) was used for inhibition of the caveolin endocytosis pathway [39]. Cells were then treated for 2 h at 37 °C with 2 mL medium at pH 6 containing fluorescent LNC modified with polymer P5, at a concentration of 250 µg/mL. To discriminate active from passive internalization, cells were treated with fluorescent LNC and incubated 2 h at 4 °C. Cells were then washed with PBS, collected after trypsinization and washed twice with PBS. To discriminate adsorbed and internalized LNC, cells were resuspended in 2.5 mg/mL trypan blue to quench extracellular fluorescence [40] and exclude dead cells. Analyses were performed with a BD FACSCanto™ II flow cytometer (BD Bioscience). Experiments were repeated five times with three replicates per experiment.

### 2.6. Confocal Microscopy

Confocal imaging of internalized cells was performed using a confocal laser microscope (TCS SP8, Leica, Switzerland), equipped with a 50 mW diode laser. SK-Mel 28 melanoma cells were seeded onto 8-well polymer μ-slides (ibidi, GmbH, France) at the density of 25 × 10^4^ cells/well and incubated for 24 h. The cells were then treated for 2 h with 300 µL of the medium at different pH (7.4, 6.8, 6.5, 6) with fluorescent LNC at a concentration of 250 µg/mL. Cells were washed with PBS and fixed in 4% paraformaldehyde for 20 min at 4 °C. Cells were then washed with PBS, and nuclei were stained with 4′,6-diamidino-2-phenylindole 3 µM DAPI (Sigma-Aldrich). Finally, cells were washed with PBS, and Fluoromount™ aqueous mounting medium (Sigma-Aldrich) was added to the slide. All images were collected with a Leica TCS SP8 AOBS confocal laser scanning microscope (Leica Microsystems, Wetzlar, Germany) equipped with an HC PL APO CS2 63x/NA 1.40 oil objective with 2x numerical zoom and 40x/NA 1.30 oil objective and gateable hybrid detectors (GaAsP). Images were acquired in the format 1024 × 1024 pixels, the bit depth of 8, and a scan speed of 400 Hz. DAPI and DiA were excited with a 405 nm diode laser (50 mW) and the 488 nm line from an argon laser (40 mW), respectively. Z-series optical sections were collected with a step size of 1 µm using a Super Z Galvo Type H stage and were displayed as maximum z-projections using the LAS X software.

### 2.7. Complement Activation

Complement activation was determined by measuring the lytic capacity of normal human serum after exposure to the LNC, according to Passirani et al. [41]. Briefly, normal human serum (NHS) (provided by the Etablissement Français du Sang (Angers, France)) was diluted in Veronal-buffered saline containing 0.15 mM Ca^2+^ and 0.5 mM Mg^2+^ (VBS^++^) and incubated for 1 h at 37 °C with an increased concentration of LNC. The suspension was then diluted in VBS^++^ (1/25 *v/v*) and incubated for 45 min at 37 °C with sheep erythrocytes (Labor Dr. Merk) sensitized by rabbit anti-sheep erythrocyte antibodies (Eurobio, Les Ulis, France). The suspension was centrifuged at 800× *g* for 10 min. The light absorption of the supernatant was then read at 405 nm with a microplate reader (Multiskan Ascent, Labsystems SA, Les Ulis, France). The amount of serum able to lyse 50% of sensitized erythrocytes (CH50) was calculated for each sample using the formula:$$consumption\;\left(\%\right)=\frac{\left(CH50_{sample}-CH50_{control}\right)\times 100}{CH50_{control}}$$

To compare the different modified LNC, the complement consumption was plotted as a function of the surface area. The surface area was calculated using the formula described previously [41].

### 2.8. Statistical Analysis

For the statistical analysis, the results were analyzed using a Kruskal–Wallis test followed by a Dunn’s post hoc test with a Hochberg correction using the R software (R Foundation, Austria) with the PMCR package [42]. The level of significance was set at *p* < 0.05.

## 3. Results

### 3.1. Polymer Post-Insertion and Switch Charge Capacities of Modified LNC

The synthesis of a series of hydrophilic NVP and Vim-based copolymers (data not shown) bearing a hydrophobic aliphatic chain in α-position (C_18_H_37_-P(NVP-*co*-Vim)) has been performed with different compositions to potentially modulate the surface charge of the LNC (Table 1, entries P2–P5). For the sake of comparison, we also prepared a PNVP homopolymer (Table 1, entry P1). After post-insertion of these polymers at the LNC surface, the size and zeta potential of the modified LNC were analyzed at different pH as reported in Figure 2. For all conditions (different pH and post-inserted polymers), the PDI observed was below 0.05 (Figure 2C).

Post-inserted LNC with polymers P3, P4 and P5 showed significant differences in their mean hydrodynamic diameters, compared to unmodified LNC (BLK) at the same pH (*p* value was below 0.05 at pH 7.4 for LNC post-P3 and below 0.01 at another pH for LNC post-P4 and P5) (Figure 2A). The diameter of this unmodified LNC was around 52 nm, while all post-inserted objects showed a higher mean hydrodynamic diameter (from 55 to 62 nm). Moreover, this post-insertion was stable over 4 weeks, and no impact of the pH was observed in terms of size and zeta potential over the studied period (Appendix A). Under neutral conditions (pH 7.4), the zeta potential of the modified LNC was nearly neutral (Figure 2B). Surface charges of LNC modified with pH-responsive polymers were directly linked to the pH: decreased pH led to an increased surface charge of pH-responsive LNC up to 17 mV for LNC post-P5 (pH 6). This zeta potential change was confirmed for all LNC modified with copolymers (P2, P3, P4 and P5). Compared to the pH 7.4 condition, they all showed a significant difference (*p* < 0.05) at pH 6.5 and pH 6 with an increase of around 12 mV between pH 7.4 and 6.

### 3.2. Buffering Effect of pH-Responsive LNC

The ability of LNC to escape endosomes efficiently was evaluated by studying the buffering capacities of each polymer and modified LNC through acid–base titration, as reported in Figure 3A,B. Buffering capacity of the polymers and modified LNC was determined by establishing the amount of HCl needed to go from pH 7.4 to 5.1. BLK LNC, LNC post-P1 and polymer P1 solution reached pH 3 very quickly. For pH-responsive LNC (post-inserted with P2, P3, P4, P5) and corresponding polymers, the addition of HCl solution decreased the pH value progressively. Moreover, the slope of the curves changed, with a slow decrease in the range of pH 4.5~7. Differences were observed between pH-responsive polymers: for example, polymer P5 showed a higher buffering capacity compared to polymer P4. This difference was directly correlated to the amount of Vim units in the polymers.

### 3.3. Stealth Properties of LNC

Complement consumption for each LNC was studied, and it increased with LNC concentration (Figure 4). BLK LNC did not present any activation, whatever the surface studied, as previously observed [43]. LNC post-inserted with polymers P1 to P4 showed a similar evolution, slightly higher than that of BLK LNC. For the same contact surface (for example, at 1740 cm^2^/mL), LNC post-P5 presented a higher consumption of CH50 units: 44% compared to 20~25% for all other post-inserted LNC and 11% for BLK LNC.

### 3.4. Impact of Polymers and Modified LNC on Cell Viability

Due to the heterogeneity between human and murine cell lines, the impact of polymers on cell viability was evaluated on B16F10, a murine melanoma cell line, and on SK-Mel 28 cells, a human melanoma cell line. The impact of modified LNC is reported in Figure 5A,C and the viability of cells treated by polymers alone are reported in Figure 5B,D. No cell viability reduction was observed for cells treated with polymers (Figure 5B,D). Compared to blank LNC, post-inserted LNC did not show any significant difference in cell viability for both cell lines. No significant impact on cell viability was observed at the different studied concentrations excepted at 1000 µg/mL: viability was reduced to 80% for B16F10 (Figure 5A) and 60% for SK-Mel 28 (Figure 5C).

### 3.5. Cell Uptake of pH-Responsive LNC

#### 3.5.1. pH-Dependent Cellular Uptake

Compared with BLK LNC, LNC post-P5 showed a significant increase of internalization at pH 6 and 6.5 for B16F10 cells (Figure 6A) and at pH 6, 6.5, 6.8 for SK-Mel 28 cells (Figure 6B). Furthermore, media acidification showed a negative impact on internalization in both cell lines: at pH 6, a significant reduction (compared to pH 7.4) was observed for BLK LNC and post-inserted LNC with P1, P2, P3 and P4 polymers (*p* < 0.01). Interestingly, this reduction was not observed with the LNC post-P5 (*p* > 0.05). Figure 6C shows representative images of cells after treatment with BLK LNC and LNC post-P5 in various pH conditions, and cells treated with other modified LNC are reported in Appendix A. A decrease of fluorescence linked to a decrease of pH was observed for cells treated with BLK LNC. On the contrary, cells treated with LNC post-P5 showed an increased fluorescence at pH 6.5 and 6. Orthogonal sections showed nanoparticles’ presence in the cytoplasm, characterized by green fluorescence: nanoparticles in the cell compartment indicated passage through the cell membrane. Therefore, due to this impressive pH-responsive behavior of LNC post-P5 in terms of cell uptake, this modified LNC has been kept and studied more deeply for the rest of this work.

#### 3.5.2. Internalization Pathways

Taking into consideration the efficient internalization of LNC post-P5 in the cytoplasm under acid conditions, the cellular uptake mechanism of these modified objects was investigated at pH 6. Control condition received only LNC post-P5, while other conditions were pretreated with different uptake inhibitors. DAM inhibits the Na^+^/H^+^ exchanger involved in macropinocytosis [44]. MβC disturbs the formation of both caveolin-coated endocytic vesicles and clathrin-coated pits [38,45]. This inhibitor is also symptomatic of macropinocytosis, as reported previously [46]. PMA inhibits caveolin-dependent endocytosis [39], and chlorpromazine inhibits clathrin-dependent endocytosis by blocking the formation of membrane invaginations [47]. Additionally, to discriminate active from passive pathways, LNC post-P5 treated cells were incubated at 4 °C. Control LNC uptake was considered as 100% of internalization. Results are reported in Figure 7A for B16F10 and Figure 7B for SK-Mel 28 cells. Both cell lines showed a significant reduction of internalization at 4 °C (*p* < 0.001). Cellular uptake of LNC post-P5 in B16F10 was significantly decreased (*p* < 0.05) by approximately 50% after pretreatments with DAM and MβC. For SK-Mel 28, cellular uptake was significantly reduced (*p* < 0.05) by 60% and 80% after pretreatments with DAM and PMA, respectively. At 4 °C, energy-dependent pathways, such as endocytosis and pinocytosis are blocked. Inhibition of cell uptake observed at this temperature highlighted active internalization of LNC post-P5 in these melanoma cell lines, rather than diffusion across the cell membrane.

## 4. Discussion

Interestingly, this study allowed developing new nanovectors sensitive to the variations in pH described at the tumor level. This project opens up interesting prospects for improving the targeting of metastatic melanoma cells resistant to various therapies.

Surface modification of lipid nanocapsules is usually done by post-insertion of lipid polymers composed of double carbon chains, for example, 1,2-dimyristoyl-sn-glycero-3-phosphoethanolamine (DMPE) or 1,2-distearoyl-sn-glycero-3-phosphoethanolamine (DSPE) [48,49]. In this study, the surface modification of lipid nanocapsules (BLK LNC), already composed of 15 units of PEG at the surface (Kolliphor^®^ HS 15) (see blue corona on graphical abstract LNC representation), was realized with the C_18_H_37_-P(NVP-co-Vim) copolymers reported in Table 1 and used as post-insertion agents thanks to a C_18_ mono carbon chain. The post-insertion of polymers at the LNC surface was confirmed by an increase in their hydrodynamic diameter and a change in their surface charge. Surface charge is a key parameter for stability, circulation time and cellular uptake. As expected, the zeta potential of the pH-responsive LNC was driven by Vim protonation and increased according to the Vim content in the polymers (BLK LNC, LNC post-P1 < LNC post-P2 < LNC post-P3 < LNC post-P4 < LNC post-P5). With a pKa value of our polymers around 6, following literature [50,51], and an extracellular tumor pH of 6.5 or below [52], Vim groups are protonated in such an acidic environment, leading to the observed increase of LNC zeta potential.

Furthermore, the protonation of Vim groups in an acidic environment can bring buffering property to LNC. The buffering capacity of pH-responsive polymers plays a key role in preventing degradation of encapsulated therapeutics, such as nucleic acid, for example, by escaping the endosome through the “proton sponge effect” [53]. Some studies reported P(Vim) as a polycationic polymer able to efficiently deliver genetic material thanks to this buffering property [54,55]. Moreover, siRNA or ferrocifen loaded LNC developed in previous works have already shown a relevant therapeutic efficacy on melanoma tumor progression [56,57]: surface modification of these nanocarriers with C18H37-P(NVP-co-Vim) copolymers could, therefore, allow them to escape the endosome and would be suitable for this delivery application.

Once intravenously injected, a possible neutralization of our LNC by the immune system of the receiving host or by natural killer-associated microvesicles could occur. The accelerated blood clearance (ABC) phenomenon is one of the pharmacokinetic consequences of this immune reaction, as observed with pegylated nanocarriers. This phenomenon was not observed with PNVP that did not potentiate any IgM production after a second intravenous injection [58,59], making PNVP a promising alternative to PEG. In this study, to predict the behavior of the modified LNC, their ability to escape complement protein opsonization was evaluated by CH50 assay as it can mediate interaction with macrophages, resulting in possible elimination of the nanocarriers from the bloodstream. This method determines the residual complement activity after incubation with nanoparticles, a high complement consumption proving its activation. The difference in complement consumption observed between BLK LNC, and post-inserted LNC can be first explained by the smaller size of BLK LNC than post-inserted LNC. Smaller objects exhibit increased surface radius, which limits protein opsonization onto the surface of nanocarriers. Other studies have shown that bigger LNC led to an increase in complement activation [43]. Compared to all other modified LNC, LNC post-P5 appeared to induce more complement activation. Interestingly, LNC post-P5 has the same hydrodynamic diameter as LNC post-P4. However, at the same pH, these nanoobjects showed different zeta potential. In this study, Veronal-buffered saline at pH 7.4 was used. At such pH, the zeta potential of LNC post-P5 was slightly positive (~+2 mV), and the other modified LNC were neutral and negatively charged. It has been shown that nanoparticles with positively charged surfaces induced complement activation [60]. Thus, the slight increase in the zeta potential of LNC post-P5 could explain the reduced protection against complement activation compared to the other post-inserted LNC. However, compared with the literature [43], complement activation of LNC post-P5 remains weak and should, therefore, not be considered as a complement activator.

The impact on cell viability of LNC decorated with our polymers was evaluated on two cell lines and did not show any decrease in cell viability. These findings could be explained by the maximum molar mass of Vim used in the copolymers corresponding to P5 (1400 g·mol^−1^) (Table 1). This molar mass is much lower than what was reported by Velasco et al., who found an IC_50_ of 0.57 mg/mL for fibroblasts treated with 200,000 g·mol^−1^ PVim [50]. Furthermore, our data confirm previous studies that suggest that PNVP nanoparticles were nontoxic and well-tolerated by animals [61]. Cellular uptake of LNC by human SK-Mel 28 and murine B16F10 melanoma cell lines was studied in various pH media values to investigate the effect on cell internalization by increasing the charged surface. The reduced internalization caused by media acidification confirmed previous data from Gündel et al., who observed a reduced internalization of uncharged dextran polymer in AT1 rat prostate cells under acidic conditions [62]. These results suggest that cell uptake of nanocarriers can be affected by the extracellular pH of tumors and lead to a reduction of their therapeutic effect. Interestingly, LNC post-P5 did not show this reduction of internalization caused by pH. On the contrary, the increase of zeta potential of LNC post-P5, caused by the acidic environment, improved their cellular uptake. As expected, when the pH decreased, the increase of LNC post-P5 surface charge contributed to enhancing the affinity between the negatively charged cell membrane and the positively charged LNC. These data confirm the interest of these nanovectors in targeting tumor cells, and particularly those located in the tumor where the pH values are reduced.

These results were confirmed by confocal microscopy. Due to the highest pH-responsive behavior of LNC post-P5 compared to other post inserted LNC, we investigated the internalization mechanism of such modified LNC. Interestingly, B16F10 and SK-Mel 28 showed different mechanism pathways. The B16F10 cell line showed a reduction of cell uptake with DAM and MβC, suggesting that LNC post-P5 used macropinocytosis, whereas cell uptake of LNC post-P5 by SK-Mel 28 occurred through a balance between macropinocytosis and caveolin-dependent endocytosis. Cellular uptake pathways can directly impact the fate of nanocarriers in the cytoplasm. Compared to macropinocytosis and clathrin-dependent endocytosis, caveolae-mediated endocytosis is a preferred pathway as the caveosome cellular compartment is not subject to pH and enzyme degradation as observed in endosomes and lysosomes. Cell uptake pathways can differ with the cell line. The “proton sponge effect” of these modified LNC should, therefore, provide the capacity to escape endosomal degradation if cellular uptake occurs through macropinocytosis and clathrin-dependent endocytosis pathways.

The MDR phenotype on the membrane level is characterized by three main features; reduction of the transmembrane diffusion of the hydrophobic drugs, efflux of the membrane entrapped drug molecules by the ABC transporters and impairment of the endocytic function. Our data suggest that, due to our formulation strategy, pH-sensitive nanovectors would be able to reverse MDR phenotype to improve cancer treatment. Furthermore, as previously studied by our group [57,63,64,65], different post-insertion strategies applied to LNC offer prospects for obtaining conclusive results about the effective ability of these nanoparticles to be loaded with the substances of interest, such as DNA, RNA or other drugs as ferrocifens.

## 5. Conclusions

The surface coating of nanomedicines by hydrophilic polymers is a common modification used to provide prolonged blood residence. With the increased presence of anti-PEG antibodies in the general human population [66], alternatives to PEG should be considered as a necessity to develop new classes of stealth nanocarriers. In this study, an innovative hydrophilic and pH-responsive copolymer bearing a lipophilic chain-end has been used as an alternative to DSPE-PEG for decoration of LNC, providing them a stealth behavior in physiological conditions and responsiveness to the acidic tumor environment. These modified LNC exhibited good stability over several weeks. In physiological conditions (pH 7.4), post-inserted LNC showed neutral zeta potential, whereas, under acidic conditions, the zeta potential increased, depending on the polymer composition and on the pH. LNC modified with polymer P5 (C18H37-P(NVP21-co-Vim15) showed impressively in vitro pH-responsive properties, characterized by an enhanced cellular uptake under acidic conditions, making them very promising carriers to target melanoma whose extracellular acidity has been correlated to highly metastatic and invasive tumors [67,68]. In conclusion, surface modification by polymer P5 would lead to an increased biological effect by not only protecting the nanocarrier from opsonization by complement activation but also by a potential increased cellular uptake in the acidic tumor microenvironment.

## Figures and Tables

**Figure 1 cancers-13-02028-f001:**
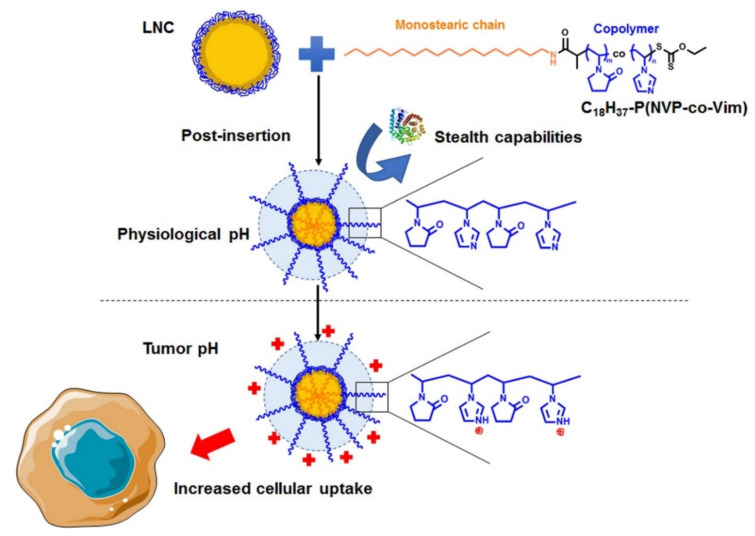
Decoration of LNC by C_18_H_37_-P(NVP-co-Vim) copolymers.

**Figure 2 cancers-13-02028-f002:**
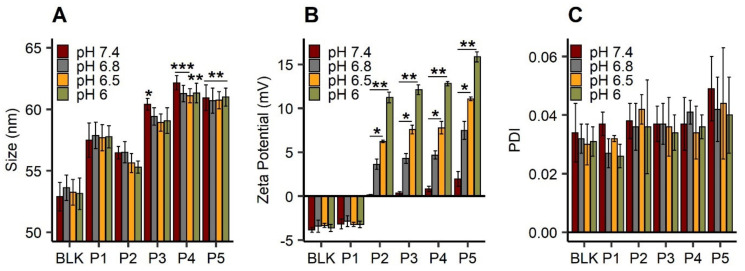
Physicochemical characterization of blank LNC (“BLK”) and LNC post-inserted by C_18_H_37_-PNVP_49_ (“P1”), C_18_H_37_-P(NVP_15_-co-Vim_5_) (“P2”), C_18_H_37_-P(NVP_22_-co-Vim_8_) (“P3”), C_18_H_37_-P(NVP_35_-co-Vim_10_) (“P4”) and C_18_H_37_-P(NVP_21_-co-Vim_15_) (“P5”): hydrodynamic diameter (nm) (**A**), zeta potential (mV) (**B**) and polydispersity index (PDI) (**C**) at different pH: 7.4, 6.8 and 6.5. Results (*n* = 4) are expressed as mean measure ± standard deviation. Statistical analysis was performed with Kruskal–Wallis, post hoc Dunn’s, correction Hochberg. For statistical analysis of size, BLK LNC was used as reference, *** *p* < 0.001, ** *p* < 0.01, * *p* < 0.05.

**Figure 3 cancers-13-02028-f003:**
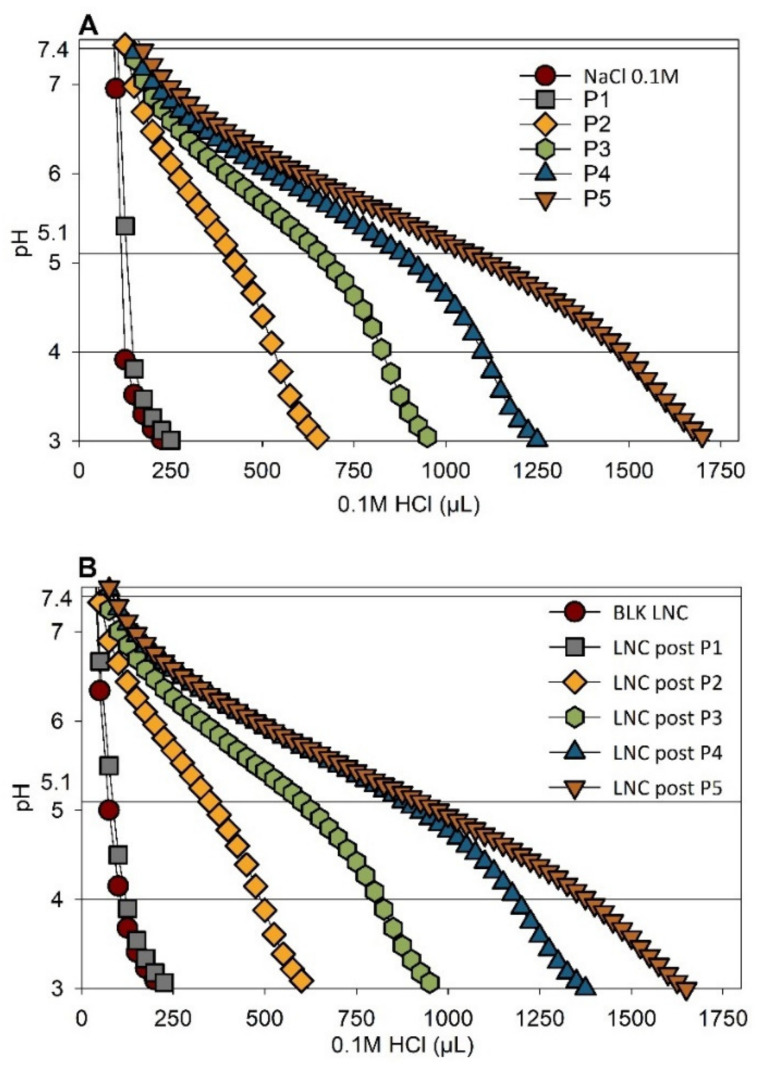
Endosome buffering effect. Acid–base titration of (**A**) polymer solution at a concentration of 0.1 mM and (**B**) LNC post inserted. Blank LNC (“BLK LNC”) and modified LNC with polymer C_18_H_37_-PNVP_49_ (“P1”), C_18_H_37_-P(NVP_15_-co-Vim_5_) (“P2”), C_18_H_37_-P(NVP_22_-co-Vim_8_) (“P3”), C_18_H_37_-P(NVP_35_-co-Vim_10_) (“P4”) and C_18_H_37_-P(NVP_21_-co-Vim_15_) (“P5”).

**Figure 4 cancers-13-02028-f004:**
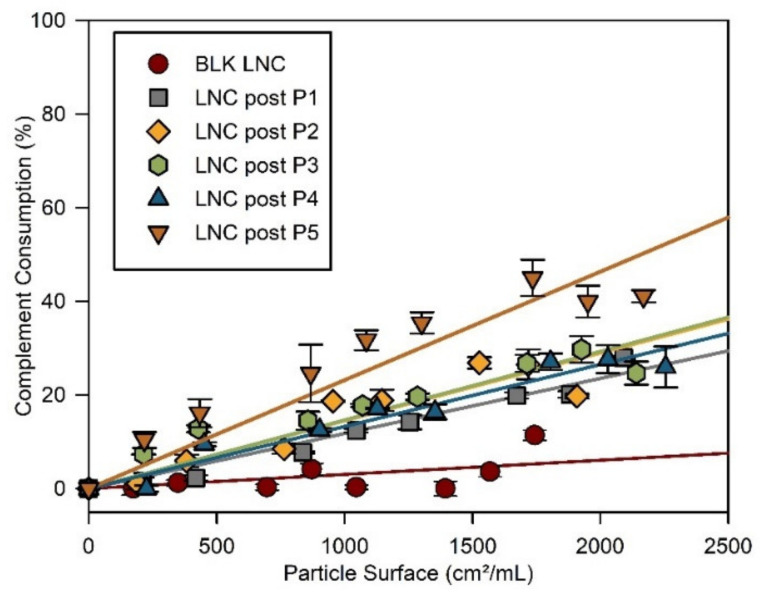
Complement consumption at 37 °C of blank LNC (“BLK LNC”) (red circle) and LNC post inserted by C_18_H_37_-PNVP_49_ (“LNC post-P1”), C_18_H_37_-P(NVP_15_-co-Vim_5_) (“LNC post-P2”), C_18_H_37_-P(NVP_22_-co-Vim_8_) (“LNC post-P3”), C_18_H_37_-P(NVP_35_-co-Vim_10_) (“LNC post-P4”) and C_18_H_37_-P(NVP_21_-co-Vim_15_) (“LNC post-P5”).

**Figure 5 cancers-13-02028-f005:**
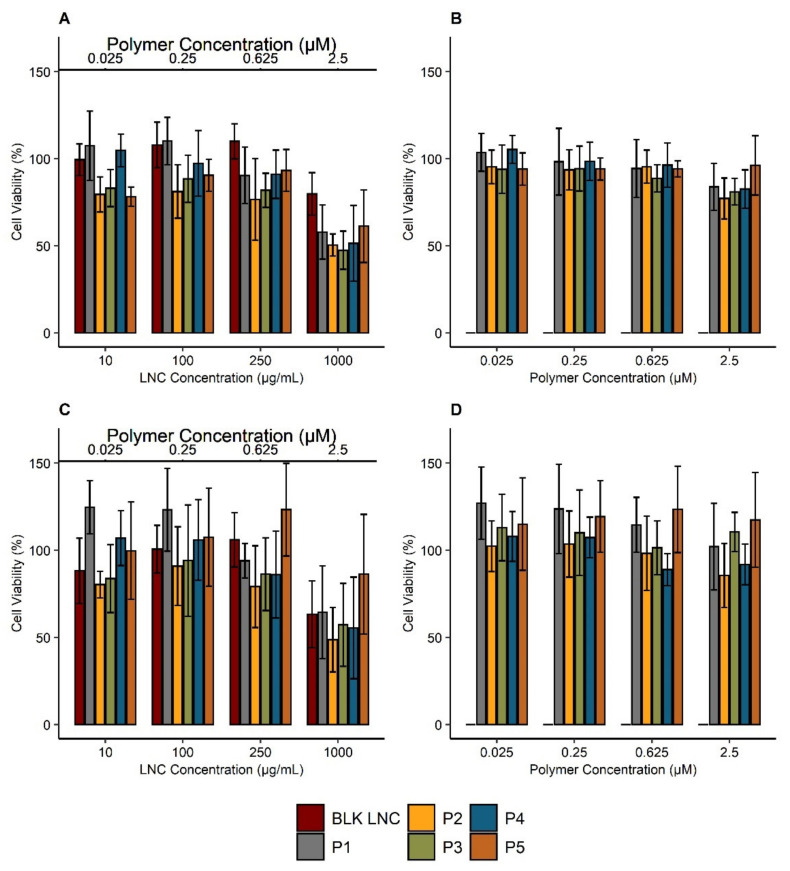
Cell viability of B16F10 and SK-Mel 28 cell lines treated with polymers and modified LNC. Cells were incubated with blank LNC (“BLK”) and LNC post-inserted by C_18_H_37_-PNVP_49_ (“P1”), C_18_H_37_-P(NVP_15_-co-Vim_5_) (“P2”), C_18_H_37_-P(NVP_22_-co-Vim_8_) (“P3”), C_18_H_37_-P(NVP_35_-co-Vim_10_) (“P4”) and C_18_H_37_-P(NVP_21_-co-Vim_15_) (“P5”) (concentration range: 10–1000 μg/mL) for 24 h: B16F10 (**A**), SK-Mel 28 (**C**). In addition, to evaluate the impact of polymers alone, cells were treated with polymers at concentrations corresponding to the amount of post-inserted polymers into LNC (0.025 µM–2.5 µM): B16F10 (**B**), SK-Mel 28 (**D**). Cell viability was then measured by resazurin reduction assay. Results (*n* = 6) are expressed as the means ± SD.

**Figure 6 cancers-13-02028-f006:**
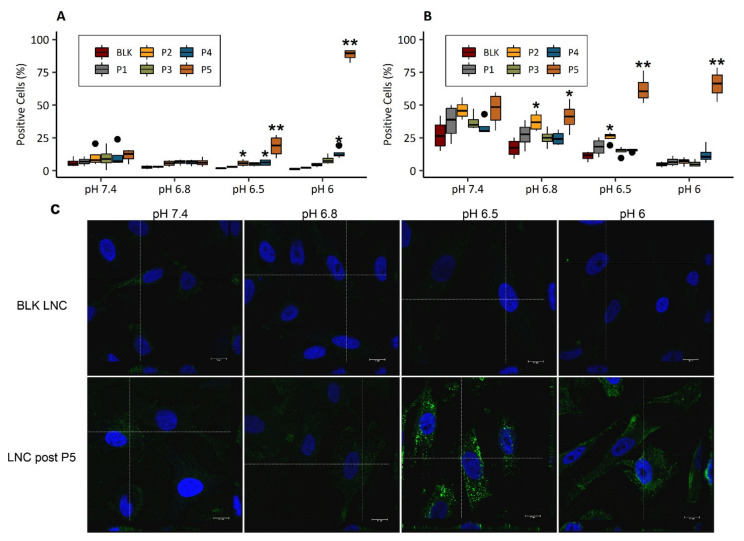
LNC internalization in B16F10 (**A**) and in SK-Mel 28 cells (**B**,**C**) assessed by FACS analysis (**A**,**B**) and by confocal imaging (**C**). Cells were treated in various pH media (7.4, 6.8, 6.5, 6) with 250 µg/mL of fluorescent LNC for 2 h. For cell uptake assessed by FACS analysis (**A**,**B**), data are expressed as a boxplot. Each box represents 50% of the distribution (interquartile range: 25th percentile–75th percentile). Median is marked as a black line in each box. Extending lines from boxes show minimum and maximum values. Outlier data are plot as black circles. Statistical analysis was performed on results (*n* = 4) with Kruskal–Wallis, post hoc Dunn’s, correction Hochberg, blank LNC (“BLK LNC”) was used as control, ** *p* < 0.01, * *p* < 0.05. For confocal imaging (**C**) of SK-Mel28 cells, after 2 h incubation with blank LNC (“BLK LNC”) and LNC post inserted by C_18_H_37_-P(NVP_21_-co-Vim_15_) (“LNC post-P5”), at pH 7.4, 6.8, 6.5 and 6. The cell nucleus was stained with DAPI (in blue), and the green signal comes from the fluorescent LNC. Scale bars correspond to 10 µm. The objective used: 63x/NA 1.40 oil with 2x numerical zoom. Orthogonal sections are localized by the dotted lines.

**Figure 7 cancers-13-02028-f007:**
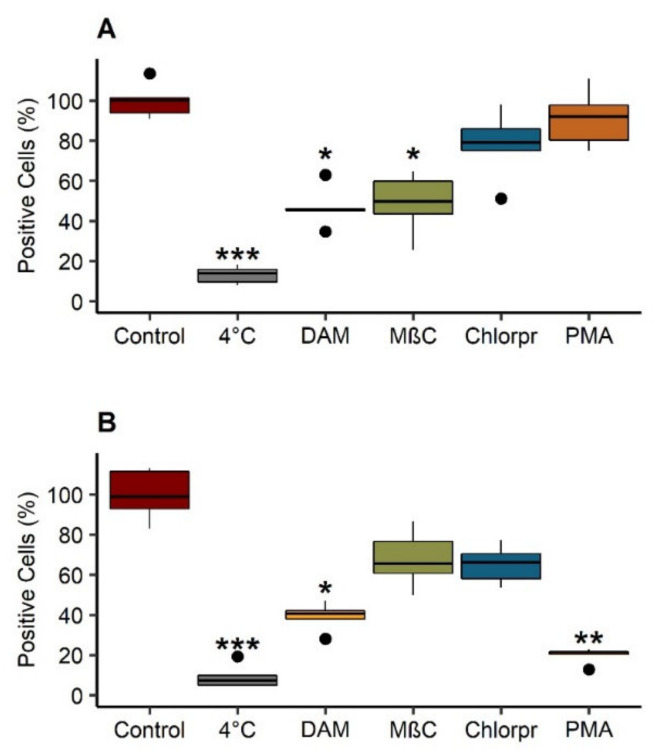
Internalization pathways of pH-sensitive LNC in B16F10 (**A**) and SK-Mel 28 (**B**) at pH 6. Cells were pretreated for 30 min with 1 mmol/L 5-(*N*,*N*-dimethyl)amiloride hydrochloride (DAM), 10 mg/mL methyl-β-cyclodextrin (MβC), 50 µmol/L chlorpromazine (chlorpr.) and 10 µg/mL phorbol 12 myristate (PMA). Then, they were incubated with 250 µg/mL of LNC post inserted by C_18_H_37_-P(NVP_21_-co-Vim_15_) (“LNC post-P5”), fluorescent LNC for 2 h in pH 6 medium, at 37 °C. Control: untreated with inhibitors. 4 °C: untreated with inhibitors and incubated with fluorescent LNC for 2 h, at 4 °C. The percentage of fluorescent cells was determined comparatively with control. Each box represents 50% of the distribution (interquartile range: 25th percentile–75th percentile). Median is marked as a black line in each box. Extending lines from boxes show minimum and maximum values. Outlier data are plot as black circles. Results (*n* = 5) were analyzed with a Kruskal–Wallis test, post hoc Dunn’s, correction Hochberg. *** *p* < 0.001, ** *p* < 0.01, * *p* < 0.05.

**Table 1 cancers-13-02028-t001:** Characterization of NVP and Vim-based copolymers determined by ^1^H NMR in *N*,*N*-dimethylformamide (DMF) at 80 °C (DP: degree of polymerization; Mn: number average molecular weight).

Entry	Copolymer Composition	DP	*M*_n_ (g·mol^−1^)
P1	C_18_H_37_-PNVP_49_	49	5900
P2	C_18_H_37_-P(NVP_15-_co-Vim_5_)	20	2600
P3	C_18_H_37_-P(NVP_22_-co-Vim_8_)	30	3600
P4	C_18_H_37_-P(NVP_35_-co-Vim_10_)	45	5300
P5	C_18_H_37_-P(NVP_21_-co-Vim_15_)	36	4200

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
