# Peer review of "pH-Responsive Lipid Nanocapsules: A Promising Strategy for Improved Resistant Melanoma Cell Internalization"

_cancers, 2021, doi:10.3390/cancers13092028_

Round 1

Reviewer 1 Report

This is an interesting article about the use of lipid nanoparticles as vector for a future use of treatment for metastatic melanoma. Just few and minor comments for the discussion.

In the discussion you should also speculate a possible neutralization by the immune system of the receiving host. Besides, another possible neutralization could be associated with natural killer associated microvescicles. Finally, another possible speculation, could be the fact that there are several types of melanoma, such as the ones arose on sun-exposed and not sun-exposed sites with different distribution of Vitamin D.  In this regard you could add some words about the limitation of your study and new perspectives, adding also this aspect.

Author Response

This is an interesting article about the use of lipid nanoparticles as vector for a future use of treatment for metastatic melanoma. Just few and minor comments for the discussion.

The authors thank the reviewer for this encouragement and for all the suggestions to improve our manuscript.

In the discussion you should also speculate a possible neutralization by the immune system of the receiving host. Besides, another possible neutralization could be associated with natural killer associated microvesicles.

We agree with the reviewer and propose to add the following paragraph in the revised version L431: Once intravenously injected, a possible neutralization of our LNC by the immune system of the receiving host or by natural killer associated microvesicles could occur. Accelerated blood clearance (ABC) phenomenon is one of the pharmacokinetic consequence of this immune reaction, as observed with pegylated nanocarriers. This phenomenon was not observed with PNVP that did not potentiate any IgM production after a second intravenous injection [51,52], making PNVP a promising alternative to PEG.

Finally, another possible speculation, could be the fact that there are several types of melanoma, such as the ones arose on sun-exposed and not sun-exposed sites with different distribution of Vitamin D.  In this regard you could add some words about the limitation of your study and new perspectives, adding also this aspect.

We thank the reviewer for this interesting remark. The following paragraph about CSD or non-CSD melanoma was added in the revised version, in the introduction L60:

MDR is associated with a wide range of pathological changes at different cellular and tissular levels. In a previous study, we found some differences in B16F10 and SKMel28 melanoma vasculature that could potentiate therapy resistance [4]. Moreover, MDR could be associated with somatic mutations. For instance, the most recurrent ones in chronic sun induced damage (CSD) and non-CSD melanoma affect genes in key signalling pathways involved in proliferation (BRAF, NRAS and NF1), growth and metabolism (PTEN and KIT), cell identity (AT-rich interaction domain 2 (ARID2)), resistance to apoptosis (TP53), cell cycle control (cyclin- dependent kinase inhibitor 2A (CDKN2A), and replicative lifespan (telomerase reverse transcriptase (TERT)) [5].

Conversely, in melanoma, we did not find any correlation between vitamin D production and changes in pH. It seems difficult to add speculations in our manuscript without modifying the main objectives of our study.

Reviewer 2 Report

In this well-written and presented paper, the authors describe the synthesis and study carried out on modified nanocapsules to target metastatic melanoma.

While the work as a whole is interesting and well structured, there are some relevant issues that need to be addressed:

1) Most of the bibliographic references, in particular those concerning LNCs, date back to several years ago. Why?

The bibliography needs to be updated or this data needs to be discussed.

2) Authors should present preliminary experiments (or at least report and discuss literature data) to demonstrate the effective ability of these nanoparticles to be loaded with the substances of interest (DNA, RNA or other drugs).

Minor points

In the introduction section (or at the beginning of the results section) a figure could be inserted to represent in a schematic way how the different chemical compounds are distributed in the formation of the particles.

Author Response

In this well-written and presented paper, the authors describe the synthesis and study carried out on modified nanocapsules to target metastatic melanoma.

The authors thank the reviewer for this encouragement and for all the suggestions to improve our manuscript.

While the work as a whole is interesting and well structured, there are some relevant issues that need to be addressed:

1) Most of the bibliographic references, in particular those concerning LNCs, date back to several years ago. Why?

The bibliography needs to be updated or this data needs to be discussed.

We thank the reviewer for this remark. In the introduction section, we have chosen to report the original works associated with the various described points. In addition, the strategy for producing pH-sensitive nanoparticles using polymers is original and new, which explains the low number of recent articles on this subject.

- Heurtault et al. (Heurtault et al., 2002, Pharm Res) described a novel and convenient process for the preparation of lipid nanoparticles. They identified an original structure for these particulates that resembles a hybrid between polymeric nanocapsules and liposomes;

- Garcion et al. (Garcion et al., 2006, Molecular Cancer Therapeutics) described an interaction between lipid nanoparticles and efflux pumps that resulted in an inhibition of multidrug resistance in glioma cells, both in culture and in cell implants in animal. Furthermore, they confirmed the physiological compatibility of lipid nanoparticles excipients suggesting an important step towards the development of new clinical therapeutic strategies against cancers.

- Lamprecht et al. (Lamprecht et al., 2006, Journal of Controlled Release) proposed a new approach to reverse multidrug resistance induced by lipid nanoparticles through an interaction with p-glycoprotein (P-gp). Interestingly, they found that etoposide lipid nanoparticles cell uptake was followed by a sustained drug release from lipid nanoparticles in combination with an intracellular P-gp inhibition. This process ensures a higher anticancer drug concentration inside the cancer cells.

We suggest keeping these references. In order to complete the bibliography, we added more recent references in the revised version, with 3 concerning LNC (57, 58, 59). Ref: 4; 5; 12; 17; 18; 21; 24; 51; 52; 57; 58; 59.

All references concerning our previous works will be also reviewed as mentioned in our manuscript: “Furthermore, other examples of efficacy against MDR tumors using ferrocifen-loaded LNC have been recently reviewed by our group (Idlas et al., review to be submitted)”. This review will be submitted in this same special issue.

2) Authors should present preliminary experiments (or at least report and discuss literature data) to demonstrate the effective ability of these nanoparticles to be loaded with the substances of interest (DNA, RNA or other drugs).

As we previously specified, the data reported in our manuscript are innovative. Indeed, the strategy for producing these pH-sensitive nanoparticles using new co-polymers is original and new and we do not have any data demonstrating the ability of these nanoparticles to be loaded with DNA, RNA or other drugs.

However, the efficacy of our nanoparticles post-inserted with different types of molecules (DSPE-PEG, peptide, affitin, …) has been studied in different cancer models. For example, in one of our previous studies, we have shown that post-insertion of ferrocifen-lipid nanoparticles by DSPE-PEG ensured melanoma passive-targeting after intravenous injection (Resnier et al., 2017, Pharmacol Res). These data were confirmed by a recent study conducted by our group (Topin-Ruiz et al., 2021, Int J Pharm, Ref 60). In this last one, we found that post-inserted ferrocifen lipid nanoparticles improved survival of treated mice by inducing both activation of CD8+ T lymphocytes and pro-apoptotic pathway.

Taken together, these data suggest an effective ability of the nanoparticles produced in the present study to be loaded by different agents before the copolymer post-insertion and to be pharmacologically active.

We propose to add the following paragraph in the revised version, at the end of the discussion L491:

Furthermore, as previously studied by our group [57-60], different post-insertion strategies applied to LNC offer prospects for obtaining conclusive results about the effective ability of these nanoparticles to be loaded with the substances of interest such as DNA, RNA or other drugs as ferrocifens.

Minor points

In the introduction section (or at the beginning of the results section) a figure could be inserted to represent in a schematic way how the different chemical compounds are distributed in the formation of the particles.

The authors thank the reviewer for this interesting suggestion. However, we think that the Graphical Abstract could represent the schematic way to explain how the different chemical compounds are distributed in the formation of the particles. Thus, in our opinion, the color code used in this figure allows to fully understand this organization. In “orange”, are represented the monostearic chains which are post-inserted in the core of the nanoparticles while copolymers of poly(N-vinylpyrrolidone/vinylimidazole) (in “blue”) organize themselves at the periphery of the nanoparticles. The latter are longer than the PEG chains that form an initial blue corona around the core of triglycerides (orange background) (for more information, see also Ref 14: Heurtault et al., 2002, Pharm Res).

To clarify the figure, we propose to add L411: ”see blue corona on graphical abstract LNC representation » at the beginning of the discussion.

Round 2

Reviewer 2 Report

This study is well-executed, with novel and interesting findings. In my opinion, the revised paper is appropriate for publication and criticisms are mostly minor, as outlined below:

-In the introduction section, the authors have chosen to report the original works associated with the various described points. This decision should be clearly highlighted in the text.

-In the introduction section, the authors write “Furthermore, other examples of efficacy against MDR tumors using ferrocifen-loaded LNC have been recently reviewed by our group (Idlas et al., review to be submitted)”. If the authors are unable to provide more details on this review, other examples of efficacy against MDR tumors using ferrocifen-loaded LNC should be shown.

-I apologize to the authors and editors, but I did not receive the graphical abstract and for this reason I had made that request. If possible I would like to see it.

Author Response

#Reviewer 2

This study is well-executed, with novel and interesting findings. In my opinion, the revised paper is appropriate for publication and criticisms are mostly minor, as outlined below:

The authors thank the reviewer for this encouragement and for all the suggestions to improve our manuscript.

In the introduction section, the authors have chosen to report the original works associated with the various described points. This decision should be clearly highlighted in the text.

In the introduction section, the authors write “Furthermore, other examples of efficacy against MDR tumors using ferrocifen-loaded LNC have been recently reviewed by our group (Idlas et al., review to be submitted)”. If the authors are unable to provide more details on this review, other examples of efficacy against MDR tumors using ferrocifen-loaded LNC should be shown.

In response to these two remarks, as suggested by the reviewer, we explained in the introduction of the new revised version that (L88) : “This original and groundbreaking work on LNC [15-16] has paved the way for other works showing the efficacy of ferrocifen-loaded LNC against MDR tumors [19-26], especially in malignant glioma models”. Furthermore, we added the various papers aiming ferrocifen-loaded LNC in different MDR tumor models in references 19 to 26.

I apologize to the authors and editors, but I did not receive the graphical abstract and for this reason I had made that request. If possible, I would like to see it.

We proposed to add the graphical abstract content in a new Figure 1 in the revised version and we modified the figure numbers in this version.